# Biopriming of *Pseudomonas aeruginosa* Abates Fluoride Toxicity in *Oryza sativa* L. by Restricting Fluoride Accumulation, Enhancing Antioxidative System, and Boosting Activities of Rhizospheric Enzymes

**DOI:** 10.3390/plants14081223

**Published:** 2025-04-16

**Authors:** Priya Katiyar, Neha Pandey, Boby Varghese, Keshav Kant Sahu

**Affiliations:** 1School of Studies in Biotechnology, Pt. Ravishankar Shukla University, Raipur 492 010, India; priyakatiyar080@gmail.com (P.K.); pandeyneha02@gmail.com (N.P.); 2Centre for Academic Success in Science and Engineering, University of KwaZulu-Natal, Durban 4001, South Africa; varghese@ukzn.ac.za

**Keywords:** antioxidative defense, fluoride, plant growth-promoting bacteria (PGPB), *O. sativa* L., *Pseudomonas aeruginosa*, reactive oxygen species (ROS)

## Abstract

Plant growth-promoting bacteria (PGPB) are free-living microorganisms that actively reside in the rhizosphere and affect plant growth and development. These bacteria employ their own metabolic system to fix nitrogen, solubilize phosphate, and secrete hormones to directly impact the metabolism of plants. Generating sustainable agricultural production under various environmental stresses requires a detailed understanding of mechanisms that bacteria use to promote plant growth. In the present study, *Pseudomonas aeruginosa* (MW843625), a PGP soil bacterium with a minimum inhibitory concentration (MIC) of 150 mM against fluoride (F), was isolated from agricultural fields of Chhattisgarh, India, and was assessed for remedial and PGP potential. This study concentrated on biomass accumulation, nutrient absorption, and oxidative stress tolerance in plants involving antioxidative enzymes. By determining MDA accumulation and ROS (O_2_•^−^ and H_2_O_2_) in *Oryza sativa* L. under F (50 ppm) stress, oxidative stress tolerance was assessed. The results showed that inoculation with *P. aeruginosa* enhanced the ability of *O. sativa* L. seedlings to absorb nutrients and increased the amounts of total chlorophyll (Chl), total soluble protein, and biomass. In contrast to plants cultivated under F-stress alone, those inoculated with *P. aeruginosa* along with F showed considerably reduced concentrations of F in their roots, shoots, and grains. The alleviation of deleterious effects of F-stress on plants owing to *P. aeruginosa* inoculation has been associated with improved activity/upregulation of antioxidative genes (SOD, CAT, and APX) in comparison to only F-subjected plants, which resulted in lower O_2_•^−^, H_2_O_2_, and MDA content. Additionally, it has also been reflected from our study that *P. aeruginosa* has the potential to increase the activities of soil enzymes such as urease, phosphatase, dehydrogenase, nitrate reductase, and cellulase. Accordingly, the findings of the conducted study suggest that *P. aeruginosa* can be exploited not only as an ideal candidate for bioremediation but also for enhancing soil fertility and the promotion of growth and development of *O. sativa* L. under F contamination.

## 1. Introduction

Fluorine, the 13th most abundant element in nature, primarily exists as fluoride (F) compounds within the Earth’s crust [1]. It enters the food chain through natural and geogenic sources, posing a serious threat to ecosystems [2,3]. The widespread use of groundwater has led to the extraction of water from deep borewells, which causes F contamination from mineral beds and contributes to endemic fluorosis in agricultural regions [1,4]. Weathering of F-rich minerals such as fluorite, apatite, and cryolite further exacerbates contamination, leading to F accumulation in crops, reduced growth, and lower productivity [1]. India and other Asian countries, including China, Bangladesh, and Pakistan, are among the most F-affected regions, with rice (*O. sativa* L.) cultivation being highly vulnerable to F toxicity [5]. The World Health Organization (WHO) has set a safety limit of 1.5 mg L^−1^ for F in drinking water, beyond which it causes severe fluorosis and neurological disorders. Globally, F contamination has affected bones, teeth, and soft tissues, leading to skeletal and dental fluorosis, impacting around 200 million people across 25 nations, including China, India, South Korea, Mexico, Kenya, and Nigeria [6].

Considering plants, F disrupts vital physiological and biochemical processes by impeding water and nutrient transport. It is absorbed through root vacuolar chloride channels (CLC1 and CLC2) and translocated via xylematic flow [1]. Prolonged exposure to F results in several biochemical, physiological, and molecular alterations, which may be acute or severe. Fluoride negatively impacts metabolic activities, nutrient intake, germination potential, growth and development, photosynthesis, carbohydrate metabolism, respiration, nitrogen assimilation, protein synthesis, enzyme activities, and gene expression, ultimately leading to oxidative stress due to excessive generation of reactive oxygen species (ROS) such as hydrogen peroxide (H_2_O_2_), superoxide (O_2_•^−^), and hydroxyl radical (OH) [7,8]. This oxidative stress results in membrane instability, protein degradation, and inhibition of key metabolic enzymes, including those involved in the Krebs cycle (e.g., succinate dehydrogenase, pyruvate dehydrogenase) and nitrogen assimilation (e.g., nitrate reductase) [9]. To combat oxidative damage, plants employ an antioxidant defense system comprising enzymatic components such as superoxide dismutase (SOD), catalase (CAT), and ascorbate peroxidase (APX), along with non-enzymatic antioxidants like flavonoids, ascorbic acid, and glutathione [10].

Several approaches have been explored to mitigate F toxicity in plants. Exogenous application of compounds such as spermidine, melatonin, and calcium salts has shown a promising effect in reducing F uptake and oxidative damage [7]. Melatonin, for instance, helps to restore antioxidant defenses and regulate hormone homeostasis [1]. Calcium compounds, including calcium hydroxide [Ca(OH)_2_], calcium nitrate [Ca(NO_3_)_2_], and calcium chloride (CaCl_2_), have also been incorporated as seed priming agents, enhancing seed germination, root-shoot growth, and chlorophyll stability while reducing electrolyte leakage and lipid peroxidation [4]. Sodium nitroprusside (SNP) effectively minimizes F toxicity in *Vigna radiata* L. and *Vigna. mungo* L. by reducing F uptake and enhancing antioxidant activity [11]. Additionally, biochar supplementation (50 g/kg soil) has been found to reduce F toxicity in *C. tinctorius* L. [12,13]. Pulsing silicon nanoparticles exogenously stimulate growth and assist in alleviating molecular injuries and production loss in *O. sativa* L., which resulted from F toxicity [4].

In addition to the above-mentioned ameliorative methods, bioremediation, which involves using microorganisms to detoxify contaminated environments, has gained attention as a sustainable strategy. Although slower than chemical treatments, it maintains soil fertility and ecological balance [14]. Fluoride contamination exerts selective pressure, leading to the emergence of F-resistant bacterial strains. These strains can alter F reactivity and mobility in the soil, thereby reducing plant uptake [15]. These bacteria achieve resistance through horizontal gene transfer or chromosomal modifications. Fluoride forms hydrogen fluoride (HF) upon interacting with protons and diffuses into bacterial cells as H^+^ and F^−^ ions, where it disrupts enzymatic activities [14]. However, resistant bacteria possess adaptive mechanisms such as F efflux systems, enzyme modification, and cell wall components that bind F, limiting its intracellular effects [16]. Furthermore, the bacterial cell wall is composed of phosphate, sulfhydryl, and amine groups that can efficiently reduce F intake and thus promote its adherence to bacterial surfaces [17,18,19]. Processes like bioaccumulation, biosorption, and release of siderophores or biosurfactants further aid bacterial F detoxification. Several F-tolerant strains have been identified, including *Bacillus flexus* NM25 [14], *Acinetobacter* sp. RH5 [14], *Staphylococcus lentus* [14], *Providencia vermicola* [17], *Nostoc* [18], and *Serratia* spp. [20], *P. aeruginosa* [21], which thrive in highly F-contaminated soils and offer potential for bioremediation applications.

Plant growth-promoting bacteria (PGPB) play a crucial role in sustainable agriculture by enhancing soil fertility, suppressing pathogens, and improving crop productivity [22]. These bacteria influence soil enzyme activity, regulate soil elemental cycles, and support plant growth through rhizosphere interactions [23]. Soil enzymatic activity is frequently used as a measure of overall soil microbial activity and fertility, as well as a potential indicator for assessing soil stress levels [24]. Several studies have highlighted the beneficial effects of PGPB on soil enzymatic activities. Based on their ability to increase crop yield through rhizosphere management, numerous strains of beneficial soil bacteria have been identified and are currently used to enhance agricultural sustainability and food security [25]. Several bacterial strains *viz*., *Bacillus amyloliquefaciens, Bacillus licheniformis, Bacillus megaterium, Bacillus subtilis, Pseudomonas mucilaginosus, Pseudomonas azotoformans, Pseudomonas chlororaphis, Pseudomonas fluorescens*, *Pseudomonas putida*, *Pseudomonas striata*, among others, are employed as biofertilizer products in India, China, Vietnam, Cuba, Sweden, Sri Lanka, and many other countries [26].

Among these, *Pseudomonas* spp. is a particularly well-studied rhizobacteria known for their role in plant growth promotion and disease suppression. Despite some strains being opportunistic human pathogens, many of them exhibit strong biocontrol properties against plant pathogens, making them valuable for sustainable agriculture [27,28,29]. *P. aeruginosa* in particular is highly adaptable, thriving in diverse environments, including soil-, water-, and plant-associated habitats. Its beneficial traits include hydrolytic enzyme production, bioactive metabolite synthesis, siderophore secretion, phytohormone production, nutrient solubilization, and efficient root colonization [29,30,31].

*Pseudomonas aeruginosa*, a prominent PGPB, has been identified for its potential to alleviate F stress in plants. Its mechanisms include the uptake and immobilization of F, enhancement of soil properties, and oxidative stress mitigation [32]. The selection of *P. aeruginosa* as a biopriming agent and *O. sativa* L. as a model crop for the present study was based on its isolation from F-contaminated agricultural lands of Chhattisgarh, a state where *O. sativa* L. is a staple crop and is severely affected by F toxicity. Results showed that *P. aeruginosa* (MW843625) demonstrated potential to abate the toxic effects of F in *O. sativa L.*, thereby promoting plant health by improving rhizospheric colonization, reducing F accumulation in the tissues, and enhancing antioxidant defense systems. Additionally, its dynamic turnover in soil, which encompasses growth, metabolic activity, and interactions with native microbiota, profoundly influenced soil properties and plant physiology. During its life cycle, *P. aeruginosa* contributed to nutrient cycling by solubilizing phosphates and producing siderophores, thereby improving nutrient availability and uptake in plants. Additionally, its metabolic activities have altered soil pH and organic matter content, creating a more conducive environment for plant growth. These changes not only enhance nutrient absorption but also bolster plant resilience against oxidative stress induced by F exposure. Moreover, the bacterium’s ability to form biofilms and produce secondary metabolites supports its persistence and functional stability within the soil ecosystem, ensuring sustained benefits over time [33,34]. A thorough understanding of *P. aeruginosa*’s turnover dynamics is crucial for optimizing its application in bioremediation and sustainable agriculture, particularly in F-affected regions.

## 2. Results and Discussion

### 2.1. Fluoride Resistance and Removal

The isolated bacterium identified as *P. aeruginosa* (Figure 1) exhibited resistance to relatively higher concentrations of F with a minimum inhibitory concentration (MIC) of 200 mM. The resistance of *P. aeruginosa* to F was also demonstrated by Chouhan et al. [21] and Edward Raja et al. [35,36]. This bacterium has lowered the F content in the growth medium to a significant level. The highest F removal efficiency (89.66%) was achieved at an F concentration of 60 mM, and this efficiency gradually declined with an increase in F concentration (Figure 2). The most plausible explanation for this is that at lower concentrations, the bacterium’s capacity for F absorption is not fully utilized. However, as the concentration increases, removal efficiency considerably decreases, probably as a result of saturation and a reduced availability of sorption sites on the bacterial surface [17,18]. Within the bacterial cells, various mechanisms confer tolerance, such as efflux pumps, intracellular sequestration, and detoxifying enzymes. In addition, an increase in the DNA repair mechanisms, the presence of F riboswitches, ion antiporters/transporters, and genetic mutations also play a role in enhancing F resistance in bacterial cells [37]. Other organisms like *S. lentus* and *P. vermicola* (KX941098) have also been reported to exhibit similar F removal efficiency from the growth media amended with varying F concentrations [17,18].

### 2.2. Fluoride Biosorption Potential of P. aeruginosa

Surface topography of treated and untreated *P. aeruginosa* biomass is depicted in the scanning electron microscope (SEM) images (Figure 3(Ai,Aii)). Due to the uptake of F and its adhesion to the bacterial surface, significant morphological alterations and discrete aggregates were observed. The surface of F-treated *P. aeruginosa* appeared comparatively thicker than the untreated biomass, which appeared porous. Furthermore, surface distortion and lump formation were more pronounced in the F-treated bacterial cells. Mukherjee et al., Thesai et al., and Shanker et al. [17,38,39] showed similar changes in the surface topographies of *P. vermicola* (KX926492), *B. flexus* (KX646392), and *Acinetobacter* sp. (GU566361), respectively, following F sorption. The energy-dispersive X-ray (EDX) analysis also confirmed the presence of F in the treated cells, as indicated by the F peak in the graph (Figure 3(Bi,Bii)). The presence of positive ions such as sodium, magnesium, and calcium in the bacterial cells could contribute to the uptake or sorption of F [17].

Fourier transform infrared spectroscopy has been used to characterize structural changes in bacterial cells by identifying functional groups responsible for F binding to bacterial membranes (Figure 3C). A peak at approximately 3300 cm^−1^ exhibited maximum stretching, indicating the presence of an O–H bond. Meanwhile, sharper peaks at 1100 cm^−1^, 1400 cm^−1^, 1600 cm^−1^, and 2900 cm^−1^ corresponded to primary amines (amine N-H stretch). Bands in the range of 1000–800 cm^−1^ corresponded to C–O–C and C–O–P stretching, suggesting the presence of oligosaccharides and polysaccharides in the bacterial biomass [17]. It has been found that F is absorbed by bacteria and accumulates once it enters the cell. Failure to synthesize any metabolites in metabolic pathways might explain the insignificant shifts in the transmittance of raw and F-loaded *P. aeruginosa* [17].

### 2.3. Colonization of P. aeruginosa on the Roots and F Accumulation in Plant Tissues

Fluoride-tolerant *P. aeruginosa* was investigated for its ability to colonize the *O. sativa* L. rhizosphere. The bacterium was found to be present (4.2 × 10^4^ CFU g^−1^) in the inoculated soil sample, while no colonies were detected in the control or F-treated soil, demonstrating successful colonization in the rhizospheric region. Similar results were reported by Islam et al. [32] regarding *P. aeruginosa* in *Triticum aestivum* L. This successful colonization of F-tolerant *P. aeruginosa* in the rhizosphere of *O. sativa* L. likely contributed to a reduced F uptake in the plant tissues [40,41,42]. As a result, F accumulation in plant tissues was significantly reduced in inoculated plants, with decreases of 33% in the roots, 35% in the shoots, 47% in the leaves, and 74% in the grains. In contrast, F-treated seedlings accrued 30 ± 3.65, 22 ± 2.75, 9 ± 1.54, and 2.9 ± 0.12 ppm F g^−1^ DM in the roots, shoots, leaves, and grains, respectively (Table 1). Roots, being in direct contact with the soil, accumulated significantly more F, followed by shoots, leaves, and grains. Such an alarming increase in xenobiotic levels in crops poses a serious risk to human health. Similar decreases in the accumulation of toxic elements, such as As and Zn in *V. radiata* L., *O. sativa* L., and *Zea mays* L., were, respectively, reported by Pandey and Bhatt, Pandey et al., and Jain et al. [15,43,44].

### 2.4. Plant Growth-Promoting Traits of P. aeruginosa and Its Impact on Plant Growth and Agronomic Characteristics

In vitro studies on F-resistant *P. aeruginosa* displayed multiple PGP traits, both in the presence and absence of F (Table 2). Soil bacteria play a crucial role in enhancing plant growth, even under stress conditions, by facilitating nutrient uptake, synthesizing growth hormones such as indole-3-acetic acid (IAA), solubilizing inorganic phosphorus (P), accumulating or mobilizing metals, and suppressing pathogen activity [45]. Usually, a bacterium has an immediate impact on a plant’s growth and development by employing one or more of these mechanisms [46]. One such key trait is the auxin indole-3-acetic acid (IAA), secreted by PGPB, which has a number of direct effects on plant growth and development under stressful conditions [47,48]. Bacterially derived IAA primarily enhances lateral and adventitious root development, leading to improved mineral and nutrient absorption and consequently promoting growth [49,50]. Similar findings, in concordance with the present study, have been reported for different species of *Pseudomonas* that produced IAA and supported root growth in *T. aestivum* L. [32,51], *Brassica napus* L. [47], and *Medicago truncatula* L. [52] under multiple stresses.

A significant portion of phosphorus (P) in soil remains inaccessible to plants due to its insoluble form. Approximately 1–50% of soil bacteria, including *P. aeruginosa*, can solubilize inorganic phosphates by producing organic acids, siderophores, and hydroxide ions [53,54]. These microbial activities improve P availability in the rhizosphere, enhancing phosphate uptake in *O. sativa* L. [55]. Furthermore, siderophores, which are chelating agents, enable bacteria to overcome the nutritional iron (Fe) limitation in plants due to their high specificity and affinity for binding Fe^3+^ [56]. These bacterial siderophores foster plant growth by improving nutrition and preventing phytopathogen development by sequestering Fe from the environment [57,58,59]. Additionally, bacterial ammonia production provides nitrogen for plant uptake, promoting root and shoot elongation, while hydrogen cyanide (HCN) acts as a biocontrol agent against pathogens [60,61]. It has previously been shown that a wide variety of bacterial species, including *Alcaligenes, Aeromonas, Bacillus, Pseudomonas,* and *Rhizobium*, are capable of generating ammonia and HCN in substantial proportions [62].

Under non-F conditions, inoculation *of P. aeruginosa* significantly enhanced root length (RL), shoot length (SL), fresh mass (FM), dry mass (DM), and membrane stability index (MSI) in *O. sativa* L. However, F exposure drastically reduced RL (47.82%), SL (62.56%), FM (root: 50.4%, shoot: 40.4%), DM (root: 43.8%, shoot: 38%), and MSI (31.18%) as compared to the control (Table 1). This decline was likely due to the nutrient deficiencies and F-induced membrane instability, which disrupts lipid-protein interactions and enzymatic functions like H⁺-ATPase. Many other plant species also exhibited a similar decrease in growth metrics [63,64,65]. Furthermore, F exposure renders the plasma membrane prone to lipid peroxidation and cytoskeletal instability, lowering MSI and thereby increasing electrolyte leakage from the cells [66,67,68]. Comparable F-induced growth reductions were also observed in winter wheat [69] and Tanzanian bean varieties [70]. In contrast, inoculation of *P. aeruginosa* mitigated F stress and increased RL (29%), SL (48.3%), FM (root: 30%, shoot: 23%), DM (root: 28%, shoot: 10%), and MSI (23%) compared to the F-treated seedlings (Table 1). This suggests that PGPR enhances plant growth by employing one or more mechanisms such as nitrogen fixation, phosphate solubilization, siderophore production, hormone synthesis, and promoting rhizobial or mycorrhizal symbiosis [29,32,37,71]. The hormones released by PGPB have a wide range of direct impacts on plant growth under a variety of abiotic stresses [48]. Several researchers have previously reported matching findings of plant growth in response to bacterial inoculation along with different metal stress [15,43,44,51,72].

Fluoride also negatively impacted reproductive traits in *O. sativa* L., reducing panicle length (PL) (34%), spikelets per panicle (51%), filled grain (FG) per panicle (72%), grain length (GL) (37%), grain breadth (GB) (40%), and hundred-grain weight (HGW) (23%), while increasing empty grain (EG) per panicle by 86% compared to the control. This might be due to the F accumulation in spikelets, which inhibited grain hardening [4]. However, *P. aeruginosa* biopriming significantly restored these traits to near-control levels, likely through its growth-promoting effects and reduced F uptake. Similar improvements in the grain yield have been reported in *T. aestivum* L. and *Z. mays* L. following PGPR inoculation [73,74,75,76].

### 2.5. Soil Enzymes

Activities of soil enzymes are highly sensitive to xenobiotic pollution, which indirectly reflects the ability of soil microbial populations to restore soil health [77]. The differences in soil enzyme activities across treatments were compared and shown in Table 1. Activities of soil urease, nitrate reductase, phosphatase, cellulase, and dehydrogenase were estimated before transferring the plantlets into earthen pots and were calculated as 96.42 ± 2.04 µg N-NH₄⁺ g^−1^ dw h^−1^, 1.34 ± 0.06 µg NO_2_^−^-N g^−1^ dw h^−1^, 716.2 ± 24.3 µg pNP g^−1^ dw h^−1^, 13.51 ± 1.39 µg D-Glu g^−1^ dw h^−1^, and 65.34 ± 2.06 µg TPF g^−1^ dw h^−1^, respectively. Activities of these enzymes were again calculated after crop harvest, and it was found that enzyme activities were reduced in the F-treated pots (Table 1). Several studies have documented the impact of F, usually in the form of NaF, on soil microbial activity and microbial community composition [78,79]. Wilke [80] demonstrated alterations in the chemical characteristics and microbial activities in F-contaminated humus soil and concluded that nitrite reductase, alkaline phosphatase, dehydrogenase, and arylsulfatase activities were suppressed following F addition to the soil. Reddy and Kaur [81] provided further evidence that F inhibits ATPase and soil peroxidase activity. Similarly, Rao and Pal [82] reported the detrimental effects of F on soil microorganisms, noting that high concentrations (380–1803 mg g^−1^ soil) of F impeded microbial growth and enzyme activity, as well as the breakdown of organic matter. Fluoride concentrations of 200–2000 mg g^−1^ soil have been shown to decrease denitrification, whereas F concentrations below 200 mg g^−1^ limit soil respiration and dehydrogenase activity [83].

Data gathered after P. aeruginosa treatment showed significant improvement in soil enzyme activity under both F-stressed and non-stressed conditions (Table 1). Under F-stressed and bacterium-inoculated conditions, urease activity increased by 36%, nitrate reductase activity by 57%, phosphatase activity by 14%, cellulase activity by 41.6%, and dehydrogenase activity by 50% compared to F-treated soil alone. Bacterial-assisted remediation stimulates plant growth while simultaneously improving soil quality [84]. Yu et al. [77] observed that inoculation of *Pseudomonas* sp. GHD-4 increased bacterial diversity, decreased lead (Pb) concentration, and enhanced soil enzyme activity. According to Hidri et al. [85], inoculation of *B. subtilis* enhanced soil quality by increasing the activity of soil enzymes such as urease, alkaline phosphatase, β-glucosidase, and dehydrogenase compared to the uninoculated control. Additionally, Marques et al. [86] determined that as zinc and cadmium levels increased, bacterial community diversity decreased. In contrast, *Helianthus annuus* inoculated with PGPR strains in the rhizosphere maintained a greater diversity of bacteria throughout the experimental period.

### 2.6. Total Chlorophyll and Nutrient Contents

Fluoride stress in *O. sativa* L. seedlings caused a substantial decline (50%) in the total chlorophyll (Chl) content, while *P. aeruginosa* inoculation in the stressed seedlings increased it compared to the control. As total Chl content in leaves is closely linked to photosynthetic efficiency, its reduction under abiotic stress conditions is well documented [87]. Chlorosis, inhibition of Hill activity, and decreased accumulation of photosynthetic pigments have previously been observed in seedlings irrigated with F-rich water [88,89,90]. However, inoculation of *P. aeruginosa* enhanced total Chl content by 29% compared to seedlings administered with F alone (Table 1). The higher total Chl content in response to *P. aeruginosa* treatment might be due to the enhanced uptake of minerals [54]. Consistent findings have been reported by Wang et al. [91], Islam et al. [32], Elekhtyar [92], Samaniego-Gámez et al. [93], Zhang et al. [94], and Abd El-Mageed et al. [87], who observed increased Chl content in plants treated with PGPB under various abiotic stress conditions.

Fluoride stress also adversely affected grain quality, reducing protein, total sugar, Fe, and Zn content by 1.8%, 30%, 22%, and 36%, respectively, compared to control plants (Table 1). However, inoculation with P. aeruginosa restored these nutrient levels to near-control levels in stressed plants. PGPBs have been widely reported to enhance nutrient exchange and uptake, including both macro- and micronutrients. Mechanisms such as modulation of rhizospheric pH through organic acid secretion, metal chelation by siderophores, and microbial mineralization facilitate nutrient availability to plants [75,87]. Thus, the presence of *P. aeruginosa* in F-stressed plants not only improved photosynthetic efficiency but also maintained grain nutrient composition, underscoring its potential in mitigating F toxicity and improving crop productivity.

### 2.7. Oxidative Stress and Antioxidant Defense Mechanisms

Fluoride stress led to excessive ROS accumulation in O. sativa L., increasing significantly in the roots (43% and 55%) and leaves (67% and 80%) compared to control plants (Figure 4A,B). This ROS surge, a hallmark of oxidative stress, resulted from F poisoning, which enhanced ROS production and disrupted antioxidant enzyme activities by binding to sulfhydryl groups, thereby preventing protein synthesis [76]. Similar overproduction of ROS under F stress has also been reported in *Olea europaea* L. [65], *Camellia sinensis* L. [95], *Eriobotrya japonica* L. [96], and *T. aestivum* L. [97].

However, inoculation with *P. aeruginosa* significantly mitigated this oxidative damage, reducing O_2_•^−^ and H_2_O_2_ levels by 19% and 33% in the roots and by 38% and 50% in the leaves compared to F-stressed plants (Figure 4A,B). Fluorescence microscopy confirmed that *P. aeruginosa*-treated leaves exhibited lower ROS intensity than the F-stressed tissues (Figure 5A,B), suggesting a protective role of the bioinoculant against F toxicity. This bacterial-mediated stress tolerance aligns with previous findings in *T. aestivum* L., *O. sativa* L., and *Lycopersicum esculentum* L. [4,37,97,98]. Normally, biological systems attempt to manage stress-mediated overproduction of ROS through various antioxidative defense mechanisms [99]. In addition, exogenous factors like PGPB, which can produce antioxidants in response to ROS, can also help reduce negative effects of unfavorable environmental conditions. According to studies conducted, bacteria may enhance a plant’s ability to withstand various stresses by inducing physical and chemical changes, a phenomenon known as PGPB-induced systemic tolerance, which is associated with increased levels of Chl a and b, as well as improved activity of CAT, SOD, APX, and other enzymes [100,101].

Lipid peroxidation, indicated by MDA content, increased by 45% in the roots and 48% in the leaves under F stress as compared to the control, reflecting membrane damage due to polyunsaturated fatty acid oxidation. However, *P. aeruginosa* inoculation substantially reduced MDA levels to 25% and 29% in the roots and leaves, respectively, even in the presence of F (Figure 6). These findings are consistent with studies on *H. annuus* L., *Salicornia brachiata* L., *C. sinensis* L., *Cajanus cajan* L., and *T. aestivum* L. under similar stress conditions [2,4,81,95,97,102].

Fluoride exposure also suppressed key antioxidant enzymes, with SOD, CAT, and APX activities declining in leaves by 2.4-, 4-, and 2-fold, and in roots by 1.5-, 1.9-, and 1.25-fold, respectively (Figure 7A–C). This inhibition likely resulted from F binding to active sites of these enzymes or from its interference with essential metal cofactors such as Mg, Fe, and Cu [103]. In contrast, *P. aeruginosa* significantly restored enzymatic activity, thereby reducing ROS accumulation and oxidative damage. Higher antioxidant enzyme activity is often linked to improved stress tolerance, as demonstrated in *Ricinus communis* L. and *H. annuus* L. grown in Ni-contaminated soils [104], and in *Populus deltoides* L. treated with an As-resistant *Agrobacterium radiobacter* [91]. Another study found that *P. sativum* L. inoculated with Ni- and Zn-tolerant *Rhizobium* sp. RP5 significantly increased antioxidant activities in roots and nodules and detoxified ROS to some extent [105].

RT-PCR analysis further confirmed these findings, revealing the downregulation of SOD, CAT, and APX gene expression in F-stressed plants, while *P. aeruginosa* inoculation significantly upregulated their transcription (Figure 7A–C). Expression of these genes showed comparable patterns with their spectrophotometric data, demonstrating that the transcriptional regulation of genes played a major role in regulating the activity of protective enzymes. This transcriptional regulation played a crucial role in enhancing antioxidant defense, corroborating similar observations in *V. radiata* L. and *Z. mays* L. under various abiotic stresses [18,43]. Moreover, inoculation with *B. pumilus* has been shown to enhance *O. sativa* L. growth under salinity and boron stress by upregulating antioxidant genes and improving cellular protection [106].

Overall, these results highlight that *P. aeruginosa* alleviates F toxicity by reducing ROS accumulation, lipid peroxidation, and oxidative stress while enhancing antioxidant enzyme activity and gene expression. These findings provide a promising approach for mitigating F-induced phytotoxicity through microbial inoculation.

## 3. Materials and Methods

### 3.1. Selection of Bacterial Strain

For the present study, F-tolerant bacterium (FTB), namely MT4B, was selected, which was isolated from the F-contaminated soil in the agricultural fields of Rajnandgaon District, Chhattisgarh, India (21°30′ N, 82°0′ E, 250–300 msl), following the methodology outlined by Katiyar et al. [20]. Furthermore, phylogenetic analysis of the strain was conducted by aligning the obtained sequences with related reference sequences retrieved from the GenBank database using Clustal Omega software (Ver. 1.2.0). The neighbor-joining method with MEGA X software (Ver. 10.2.0) was employed for distance analysis and phylogenetic tree construction, with bootstrap values generated from 100 replicates.

### 3.2. Fluoride Resistance and Removal Assays

The MIC of F for *P. aeruginosa* was determined by the agar dilution method. This strain was streaked onto Luria Bertani (LB) agar plates supplemented with various concentrations (5, 10, 25, 50, 75, 100, 125, 150, 200, 250, and 275 mM) of F and incubated at 35 °C for 24 h. The lowest concentration of F that inhibited isolate growth was recorded as the MIC [18].

The ability of *P. aeruginosa* to remove F was assessed by exposing it to a range of F concentrations (20, 40, 60, 80, and 100 mM) selected on the basis of MIC value. LB broth containing varying concentrations of F was seeded with the bacterial suspension and incubated for 24 h at 35 °C under shaking conditions at 100 rpm. After incubation, bacterial cultures were centrifuged at 6000 rpm for 10 min at 4 °C, and the supernatants were collected. The concentration of F was then determined using an F ion-selective electrode (Orion 9609, Thermo Scientific, Waltham, MA, USA) by mixing 10 mL of the collected supernatant with 1 mL of total ionic strength adjustment buffer (TISAB) solution (pH 5.5). The percentage of F removed by *P. aeruginosa* was then calculated using the following formula [17]:$$Percentage\;Removal=[\frac{Ci-Cf}{Cf}]\times 100$$
where, Ci and Cf are the initial and final concentrations of F in the culture media.

### 3.3. Determination of F Biosorption by P. aeruginosa

Scanning electron microscopy (JSM-IT300, JEOL, Peabody, MA, USA) was used to examine the surface characteristics of the biomass to determine changes in the shape and spatial orientation of cells after exposure to F [43]. Bacterial cells are not natural conductors; therefore, before SEM analysis, the cells were coated with platinum to increase their surface conductivity. Energy-dispersive X-ray (JEOL, Peabody, MA, USA) was carried out to determine the presence of F within the bacterial biomass [17]. Fourier transform infrared (FTIR) spectroscopy was performed following the methodology of Pandey and Bhatt [43] to analyze the interaction between F and functional groups present in untreated and F-laden biomass of *P. aeruginosa*. For the analysis, a freshly grown culture of *P. aeruginosa* was centrifuged at 6000 rpm for 15 min; the pellet was collected and dried in an oven at 60 °C. Next, the dried biomass was ground and then homogenized with potassium bromide at a 1:1 (*w*/*w*) ratio. Lastly, spectral scanning of the sample was performed using an FTIR spectrometer (Nicolet iS10, Thermo Scientific, Waltham, MA, USA) in the range 4000–400 cm^−1^.

### 3.4. Plant Growth-Promoting Activities of P. aeruginosa Under F-Stress

*P. aeruginosa* was screened for multiple PGP traits, *viz.*, IAA, ammonia, HCN, siderophore production, and solubilization of P, both in the absence and presence of F (60 mM), as this concentration exhibited the highest F removal efficiency. The quantitative analysis of IAA was performed following the method described by Gordon and Weber [107]. This strain was assessed for ammonia production by adding 0.5 mL of Nessler’s reagent to 10 mL of freshly grown cultures in peptone water, with absorbance recorded at 425 nm using a UV-Vis spectrophotometer (Lambda-25, Perkin Elmer, Waltham, MA, USA) [108]. Production of HCN was determined according to the method of Castric (1975) [109] by visualizing a yellow-to-brown color change in Whatman filter paper No. 1 soaked in 0.5% (*v*/*v*) picric acid solution, placed inside the lids of Petri plates. Siderophore production was monitored using the chrome azurol S (CAS) assay [110] by observing yellow-to-orange halos around the bacterial colonies. Likewise, phosphate-solubilizing activity of the isolate was quantitatively assayed in the National Botanical Research Institute’s phosphate growth medium (NBRIP) containing tricalcium phosphate, according to the method described by Fiske and Subbarow [111].

### 3.5. Model Plant and Experimental Design

Soil used for sowing seeds of the model species was collected from the School of Studies in Biotechnology, Pt. Ravishankar Shukla University, Raipur, Chhattisgarh, India. The collected soil was tested for the presence of F. Additionally, the soil was autoclaved to eliminate any microorganisms present to examine the effect of *P. aeruginosa* alone on plant growth. *P. aeruginosa* cultured in the LB broth was used as a bioinoculant after being diluted with 0.9% (*w*/*v*) saline. The inoculant was mixed with soil at a cell count of 10^6^ CFU g^−1^ soil [43]. Next, fresh, healthy seeds of *O. sativa* L. (Var. MTU1010) procured from the Indra Gandhi Agricultural University, Raipur, Chhattisgarh, India, were sorted and surface-sterilized with 70% (*v*/*v*) ethanol and 0.1% (*v*/*v*) sodium hypochlorite solution for 3 min, separately, followed by multiple washes with sterile distilled water (DW). Sterilized seeds were soaked in DW (as control), 50 ppm NaF solution (as stressed seeds), and 10^6^ CFU mL^−1^ of the selected strain (as bioprimed seeds) for 24 h and sown in the conditioned soil. Four experimental treatments were established: (i) seeds were sown in soil lacking both F and *P. aeruginosa* (control), (ii) seeds were sown in soil containing *P. aeruginosa* as a bioinoculant, (iii) seeds were sown in soil containing 50 ppm F, and (iv) seeds were sown in soil containing 50 ppm F and *P. aeruginosa* as a bioinoculant.

In this study, the no observed effect concentration (NOEC) and effective concentration (EC) of F were determined through a pilot experiment across a range of 10–60 ppm concentrations. Fluoride levels up to 40 ppm had no significant impact on *O. sativa L.* germination, establishing this as the NOEC. However, 50 ppm caused a 50% reduction in germination, identifying it as the EC at which F toxicity became evident. Concentrations ≥ 50 ppm led to severe phytotoxic effects, including growth inhibition and oxidative stress. Thus, 50 ppm was selected as the experimental stress condition to evaluate *P. aeruginosa*’s role in mitigating F toxicity.

Initially, seeds were sown in sterilized disposable plastic cups (each containing 30 g of soil and 10 seeds) for 25 consecutive days, with seedlings irrigated daily with 10 mL of sterile DW to ensure proper germination and growth. Thereafter, seedlings were carefully transplanted into larger earthen pots (5 kg soil capacity) filled with similarly conditioned soil. Pots were regularly irrigated with sterilized DW until plant maturation and grain ripening. All experimental sets were maintained under a standard photoperiod (10–12 h) and natural sunlight exposure (5–6 h). After full maturation and seed ripening, the roots, leaves, and grains were harvested and stored at −80 °C until the designated experiments were completed.

### 3.6. Rhizosphere Colonization by P. aeruginosa

The colonization of *P. aeruginosa* in the rhizosphere was assessed following the method of He et al. [112]. Rhizospheric soil (1 g) was collected from the pots of all four treatments, individually suspended in 10 mL of sterile DW, and kept under shaking conditions for 30 min. The resulting suspensions were used as inocula and spread onto LB agar plates supplemented with F (fluoride). Subsequently, the plates were incubated at 35 °C for 72 h, and bacterial growth was observed.

### 3.7. Determination of Soil Enzymes Activities

Soil enzyme activity was analyzed for urease, nitrate reductase, phosphatase, cellulase, and dehydrogenase before seed sowing and after crop harvest. Soil urease (EC 3.5.1.5) activity was determined by using the Shcherbakova method [113], with urea as the substrate. The concentration of released N-NH_4_^+^ following the addition of Nessler’s reagent was measured by recording absorbance at 400 nm and was expressed as µg N-NH_4_^+^ g^−1^ dw h^−1^.

Soil nitrate reductase (EC 1.7.99.4) activity was determined using the Kandeler method [114] by incubating soil with KNO_3_ as the substrate for 24 h. Activity was measured based on the amount of released NO_2_^-^ after adding a coloring reagent (sulfanilamide and 0.1 g of N-[1-naphthyl] ethylenediamine dihydrochloride) by measuring the absorbance at 520 nm and expressed as µg N-NO_2_^-^ g^−1^ dw h^−1^.

Soil cellulase (EC 3.2.1.4) activity was determined using the method of Pancholy and Rice [115] by adding sodium acetate buffer (50 mM, pH 5.5) and carboxymethylcellulose to the soil. Subsequently, the concentration of reducing sugars was measured using the Somogyi-Nelson method [116] by recording absorbance at 520 nm. Activity was expressed in the unit µg D-Glu g^−1^ dw h^−1^.

Soil phosphatase (EC 3.1.3.2) activity was determined following the method of Tabatabai and Bremner [117] with disodium *p*-nitrophenyl phosphate as the substrate. Enzyme activity was measured spectrophotometrically at 400 nm based on the amount of p-nitrophenol released and expressed as µg pNP g^−1^ dw h^−1^.

Dehydrogenase (EC 1.1.1.x) activity was measured using the method of Thaimann [118]. Soil was incubated with triphenyl tetrazolium chloride as the substrate. The content of triphenyl formazan was measured using a spectrophotometer at 546 nm, and activity was expressed as µg TPF g^−1^ dw h^−1^.

### 3.8. Assessment of Growth Attributes and Membrane Stability Index

Germination percentage, RL, and SL of mature plants were calculated as the mean of five seedlings, with three replicates. Fresh mass was estimated immediately after harvesting, while DM was recorded after drying the tissues in an oven at 70 °C for 24 h [119]. Following the method of Rady [120], MSI of plant tissues was calculated. Two sets of each treatment, in three replicates, were prepared by taking 1 g of plant tissues (leaves and roots) and immersing them in test tubes containing 10 mL of MilliQ water. The first set was kept in a water bath maintained at 40 °C for 30 min, while the second set was incubated at 100 °C for 10 min. Thereafter, the electrical conductivity of each set was monitored. The data obtained from the first set was given the code C1, while the second set was coded as C2. Finally, the following formula was used to calculate the MSI (%).$$MSI(\%)=\left(1-\frac{C1}{C2}\right)\times 100$$

### 3.9. Determination of Total Chlorophyll

Leaves (0.2 g) were collected from three randomly selected plants per treatment, homogenized in chilled 80% (*v*/*v*) acetone, and centrifuged at 5000 rpm for 10 min to pellet debris [121]. The absorbance of the supernatant was recorded at 645 and 663 nm. Total chlorophyll (Chl) content was calculated using the equation below and expressed as mg g^−1^ fresh mass (FM):$$Total\;Chl=\frac{20.2\times A645+8.02\times A663}{Fresh\;mass\;of\;leaves}$$

### 3.10. Measurement of F Content in Plant Tissues

To estimate F concentration, 1 g of tissue (roots, shoots, leaves, and grains) in triplicate was oven-dried at 70 °C for 24 h, powdered, and digested with aqua regia [concentrated HNO_3_:HCl (1:3, *v*/*v*)]. The digested samples were centrifuged at 5000 rpm for 15 min, and the supernatant was collected. The collected supernatant was mixed with total ionic strength adjustment buffer (TISAB) in a 1:1 (*v*/*v*) ratio and analyzed using an F ion-selective electrode [2].

### 3.11. Determination of Agronomical Attributes

Agronomic parameters such as panicle length (PL), spikelet count per panicle, number of filled grains (FG) and empty grains (EG), hundred-grain weight (HGW), and grain length/grain breadth (GL/GB) were estimated. Panicle length was calculated by measuring five randomly selected panicles per treatment, with three replicates. To quantify filled grains (FG) and empty grains (EG), grains were initially separated through hand threshing and winnowing, followed by manual counting. To determine HGW, the weight of 25 grains was recorded and extrapolated to calculate the HGW [4]. To determine the length and breadth of individual grains, ten grains were randomly selected from each treatment set and arranged horizontally and vertically. The length and breadth of the arranged grains were then measured manually using a scale, and the recorded data were divided by ten to obtain the average grain length and breadth [4].

### 3.12. Determination of Protein, Total Sugar, Zinc, and Iron in the Grains

Protein content in the grains was estimated using Bradford [122], and data are expressed in µg mL^−1^. Total sugar content was estimated using the method of Dubois et al. [123], and the values were expressed in µg mL^−1^. Zinc (Zn) and iron (Fe) content were quantified using the methods of Malavolta et al. [124] and Kalaimaghal and Geetha [125], respectively. Samples were prepared, and concentrations were measured in ppm using atomic absorption spectroscopy (AAS) [4].

### 3.13. Generation of ROS

The concentration of O_2_•^−^ was spectrophotometrically estimated using the method of Sangeetha et al. [126] by recording absorbance at 540 nm. Its content was calculated based on its ability to diminish nitro blue tetrazolium (NBT) and expressed as µmol min^−1^ g^−1^ FM. Hydrogen peroxide content was measured following the protocol of Velikova et al. [127]. Using an extinction coefficient of 0.28 µM^−1^ cm^−1^, its concentration was calculated and expressed as µmol g^−1^ FM.

### 3.14. Fluorescence Microscopy

The production and intensity of both O_2_•^−^ and H_2_O_2_ in the mature leaves of *O. sativa* L. were visualized using a Confocal Fluorescence Microscope 2000 LED (Leica, Germany) after incubation with dihydroethidium (DHE) and 2,7-dichlorodihydrofluorescein diacetate (H_2_DCFDA), respectively [128].

### 3.15. Lipid Peroxidation

The method of Hodges et al. [129] was adopted to assess lipid peroxidation by quantifying malondialdehyde (MDA) content. A 3 mL solution of 20% (*w*/*v*) trichloro acetic acid containing 0.5% (*w*/*v*) thiobarbituric acid was used to homogenize 0.2 g of tissues (roots and leaves). The mixture was heated at 95 °C for 30 min in a water bath and then cooled for 15 min in a freezer. The homogenate was then centrifuged at 10,000 rpm for 15 min. The absorbance of the supernatant was recorded at 540 nm, after subtracting non-specific absorbance at 600 nm. MDA content was determined using an extinction coefficient of 155 mM^−1^ cm^−1^ and expressed as mmol g^−1^ FM.

### 3.16. Enzyme Extraction

Using a chilled mortar and pestle, fresh roots and leaves (0.2 g) were individually homogenized in 2 mL of cold potassium phosphate buffer (50 mM, pH 7.0) containing 1 mM EDTA. The supernatant obtained after centrifuging the homogenate at 12,000 rpm for 15 min at 4 °C was used to assess antioxidative enzyme activity, with the exception of APX activity, which required grinding the tissues separately in a homogenization solution containing 2 mM ascorbate in addition to the other components.

### 3.17. Antioxidant Enzyme Assays

The activity of SOD (EC 1.15.1.1) was determined following the method of Marklund and Marklund [130] by measuring the percentage inhibition of pyrogallol auto-oxidation at 420 nm. In a test tube, 2.74 mL of Tris-HCl buffer (50 mM, pH 8.2) containing 1 mM of both diethylenetriaminepentaacetic acid and EDTA was prepared, followed by the addition of 0.2 mL of enzyme extract. A 60 µL aliquot of pyrogallol (0.2 mM in 10 mM HCl) was added to initiate the reaction, and after 6 min of incubation, the change in absorbance at 420 nm was recorded. SOD activity was expressed as units of SOD min^−1^ g^−1^ protein.

The method of Chance and Maehly [131] was followed to measure CAT activity (EC 1.11.1.6). The breakdown of H_2_O_2_ was estimated by monitoring the decrease in absorbance at 240 nm. The assay mixture consisted of 60 µL of enzyme extract and 37.5 mM potassium phosphate buffer (pH 6.8). A 200 µL aliquot of 60 mM H_2_O_2_ was added to initiate the enzymatic reaction, and the change in absorbance was monitored at 15 s intervals for 5 min. The extinction coefficient of 0.039 M^−1^ cm^−1^ was used to quantify CAT activity, which was expressed as µmol min^−1^ g^−1^ protein.

APX (EC 1.11.1.11) activity was determined by measuring the rate of ascorbate oxidation at 290 nm, following the method of Nakano and Asada [132]. The reaction mixture contained 2.3 mL of 0.025 M potassium phosphate buffer (pH 7.0), 10 µL of enzyme extract, 190 µL of EDTA, and 500 µL of ascorbic acid. Initial absorbance was measured at 290 nm immediately after adding 10 µL of H_2_O_2_ (0.1 M) to the assay mixture. The final absorbance was recorded after 20 min of incubation. Activity was measured using an extinction coefficient of 0.0028 M^−1^ cm^−1^ and expressed as mmol min^−1^ g^−1^ protein.

### 3.18. Gene Expression Analysis

Extraction of RNA was carried out using the hot phenol method of Verwoerd et al. [133]. After extraction, RNA quality and concentration were assessed using a Nanodrop Spectrophotometer (ND1000, Thermo Scientific, Waltham, MA, USA). Agarose gel electrophoresis (1.2%, *w*/*v*) was conducted to validate RNA quality by observing the 18S and 28S rRNA bands. cDNA was then synthesized using the Hi-cDNA Synthesis Kit (HI Media, Thane, India).

Using a set of gene-specific primers for each enzyme, a 20 µL reverse transcriptase-PCR reaction was set up. By employing a housekeeping gene, β-actin templates were normalized. Distinct primers for SOD (forward 5′-CTGATCTAGAGGGAAGTCA-3′; reverse 5′-TGTATGGGAGCATGCTACT-3′), CAT (forward 5′-CTATTGGAAGATTATCATCT-3′; reverse 5′-AGAATTCTTGATTTTTCTA-3′) and APX (forward 5′-TGGCACTCTGGGTACTTT-3′; reverse 5′-GATTTGAGGGACCATGGACT-3′) were created using software primer3 (https://www.primer3plus.com/index.html (accessed on 23 December 2017). The PCR was performed with the following cycling conditions: initial denaturation at 95 °C for 5 min, followed by 35 cycles of 95 °C for 60 s (denaturation), 49–65 °C for 45 s (annealing), and 72 °C for 60 s (extension), with a final extension at 72 °C for 5 min. The amplicons were electrophoresed on a 1.5% (*w*/*v*) agarose gel at 50 V for 40 min. Fluorescence intensity of the bands was analyzed using a Gel-Doc (Bio-Rad, Hercules, CA, USA) to determine relative gene expression levels.

### 3.19. Statistical Analysis

Data presented are mean values of three replicates ± SD and were analyzed using SPSS software (Version 20.0). One-way analysis of variance (ANOVA) was adopted followed by Duncan’s multiple range tests to measure the significant difference between the means of all the treatments at *p* < 0.05.

## 4. Conclusions and Future Prospectus

Seed biopriming with *P. aeruginosa* (MW843625), isolated from fluoride (F)-contaminated agricultural soil of Chhattisgarh, India, has demonstrated a strong ability to accumulate F and offers an effective strategy to enhance crop resilience against F stress. Analysis of its surface morphology and elemental composition confirmed F uptake. Its plant growth-promoting (PGP) traits significantly enhanced root and shoot growth, as well as biomass accumulation in *O. sativa* L. under F stress. Biopriming involves the controlled hydration of seeds in the presence of beneficial microbes, stimulating early metabolic activity before germination. It improves seed germination, seedling vigor, and protection against soil-borne pathogens. Furthermore, bioinoculation enhanced soil enzyme activity, improving soil health, reducing F absorption in plant tissues, and increasing tolerance, ultimately leading to lower F accumulation in grains. Additionally, it mitigated oxidative stress by enhancing the activity of antioxidant enzymes (SOD, CAT, APX) and improving photosynthetic pigment production. These results highlight *P. aeruginosa* as a promising biological tool for F remediation and improved crop growth in contaminated soils.

However, practical challenges include ensuring uniform application in the fields, maintaining microbial viability during storage, and achieving consistent colonization under varying environmental conditions. To address these limitations, future research should focus on developing robust formulations to enhance the shelf-life and field performance of microbial inoculants. Advancements in encapsulation technologies, such as the use of protective carriers or coatings, can protect microbial viability and enable controlled release upon sowing. Additionally, integrating seed biopriming with precision agriculture tools can ensure precise and uniform application, optimizing the benefits of this technique. Field trials across diverse agro-climatic zones are essential to validate the efficacy of *P. aeruginosa* biopriming under real-world conditions. Collaborative efforts between researchers, extension services, and farmers will be crucial for translating these innovations into practical applications, thereby promoting sustainable agricultural practices in F-affected areas.

## Figures and Tables

**Figure 1 plants-14-01223-f001:**
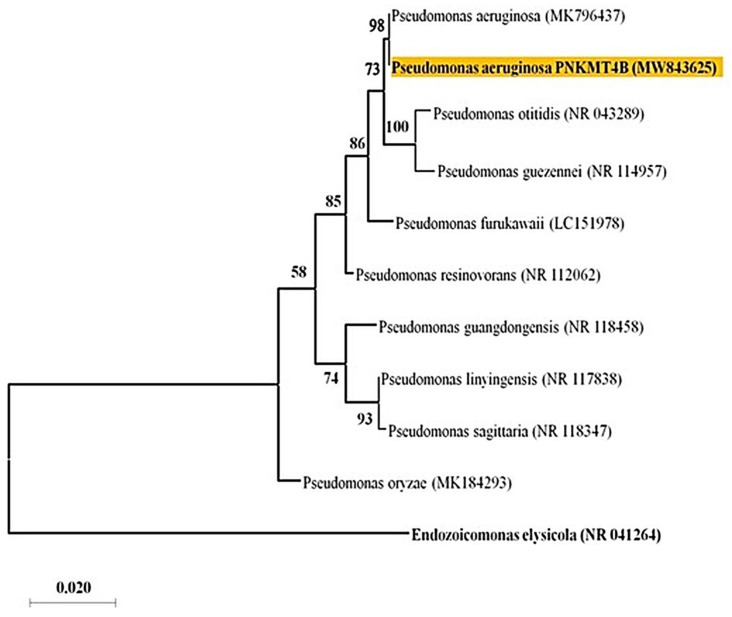
Phylogenetic tree of *P. aeruginosa* (MW843625) based on 16S rDNA.

**Figure 2 plants-14-01223-f002:**
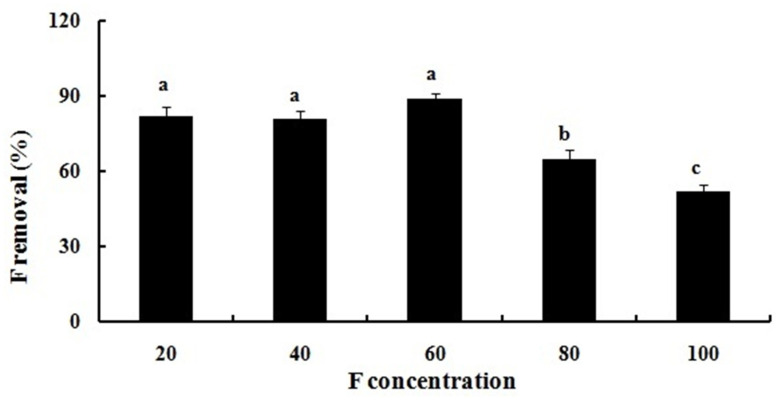
Percentage removal of fluoride by *P. aeruginosa* from growth media supplemented with different concentrations of fluoride. Different lowercase letters (a, b, c) above the bars indicate statistically significant differences among treatments at *p* < 0.05, as determined by one-way ANOVA followed by DMRT post hoc test.

**Figure 3 plants-14-01223-f003:**
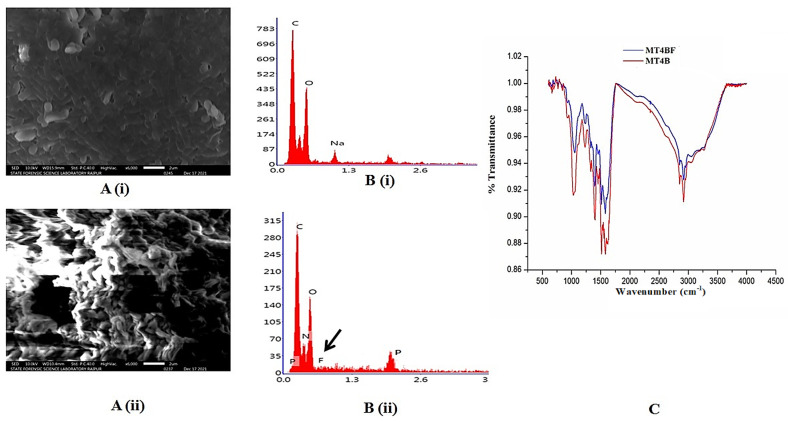
(**A**) Scanning electron microscopic images of *P. aeruginosa*: (**i**) the untreated cells having non-significant structural alterations, (**ii**) F treated cells showing severe structural distortion; (**B**) Energy dispersive X-ray analysis of *P. aeruginosa*: (**i**) absence of F peak in the untreated cells, (**ii**) presence of F peak confirms its uptake by the treated cells; (**C**) Fourier transform infrared spectra of untreated and treated *P. aeruginosa*.

**Figure 4 plants-14-01223-f004:**
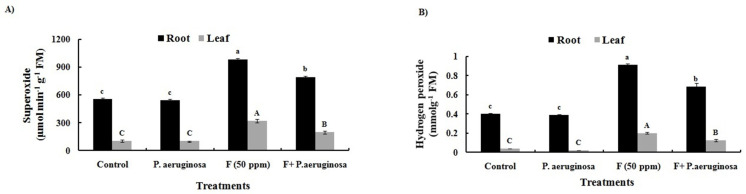
Changes in the levels of superoxide (**A**) and hydrogen peroxide (**B**) in *O. sativa* L. tissues under different treatments. All the experiments were carried out in three replicates (±SD). Different alphabets on the bars indicate significant differences (*p* < 0.05) in various treatments.

**Figure 5 plants-14-01223-f005:**
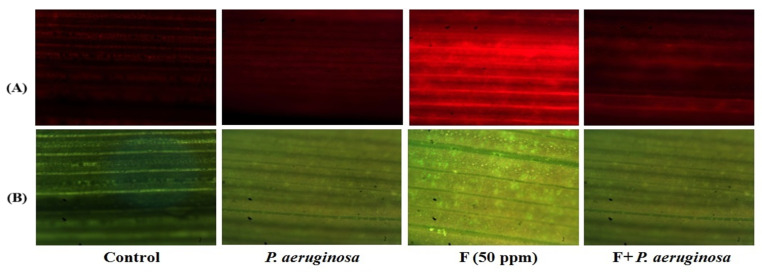
Localization of superoxide radical (**A**) and hydrogen peroxide (**B**) in the leaves of *O. sativa* L. treated with fluoride (F) and *P. aeruginosa* either alone or in combination.

**Figure 6 plants-14-01223-f006:**
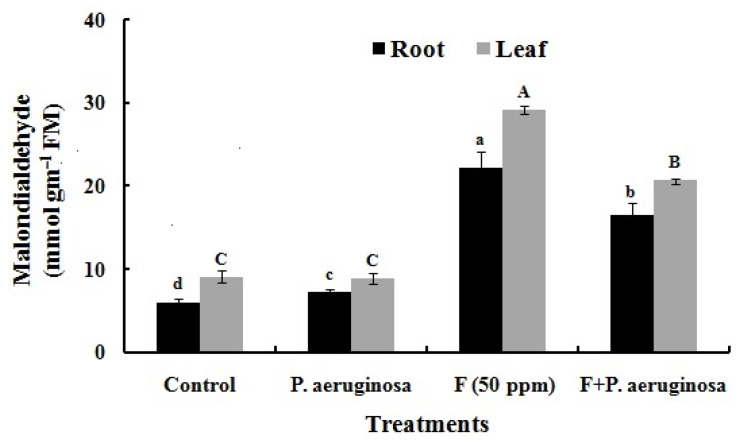
Accumulation of malondialdehyde in the roots and leaves of *O. sativa* L. of various treatments. All the experiments were carried in three replicates (±SD). Different alphabets on the bars indicate significant differences (*p* < 0.05) in various treatments.

**Figure 7 plants-14-01223-f007:**
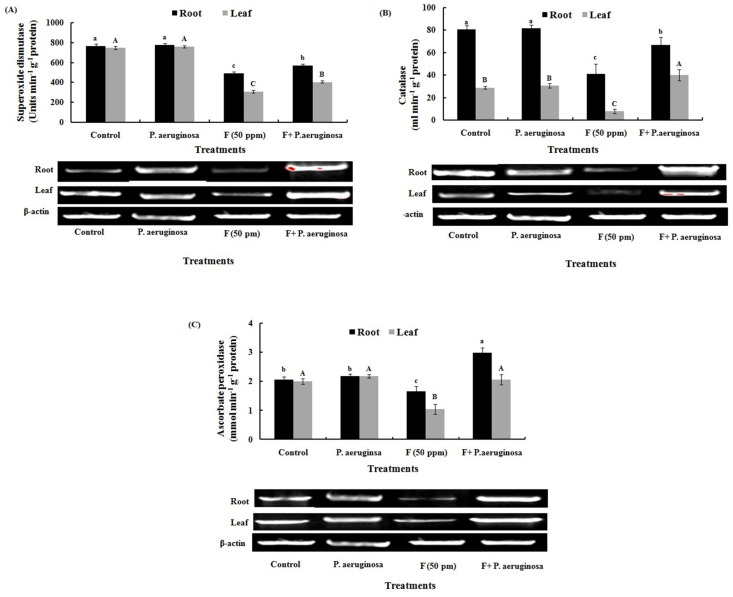
Spectrophotometric and gene expression analyses of (**A**) superoxide dismutase, (**B**) catalase, and (**C**) ascorbate peroxidase in the tissues of *O. sativa* L. under different treatments. Each bar represents mean ± SD of three independent observations. Data followed by different alphabets are statistically significant at *p* < 0.05.

**Table 1 plants-14-01223-t001:** Comparative analyses of basic physiological parameters, F accumulation in tissues, yield attributes, ROS production, and antioxidants in *O. sativa* L. seedlings grown in DW, i.e., Control, *P. aeruginosa*-treated soil, F-added soil, and *P. aeruginosa* + F-added soil. Data presented are means of three replicates ± SD. Different lowercase superscripted letters (a, b, c, d) indicate statistically significant differences among treatments at *p* < 0.05, as determined by one-way ANOVA followed by DMRT post hoc test.

(Properties)	Parameters	Control	*P. aeruginosa*	F (50 ppm)	F + *P. aeruginosa*
Soil enzyme activity (post-harvest soil samples)	Urease (µg N-NH_4_^+^ g^−1^ dw h^−1^)	98.60 ^c^ ± 1.18	129.88 ^a^ ± 1.38	67.55 ^d^ ± 0.60	105.88 ^b^ ± 0.99
	Nitrate reductase (µg N-NO_2_^−^ g^−1^ dw h^−1^)	1.826 ^c^ ± 0.063	2.401 ^a^ ± 0.078	0.487 ^d^ ± 0.06	1.139 ^b^ ± 0.067
	Phophatase (µg pNPg^−1^ dw h^−1^)	721.4 ^b^ ± 20.3	795.6 ^a^ ± 19.5	621.6 ^c^ ± 22.3	726.4 ^b^ ± 21.6
	Cellulase (µg D-Glu g^−1^dw h^−1^)	15.78 ^b^ ± 1.09	20.87 ^a^ ± 1.00	8.3 ^c^ ± 0.66	14.23 ^b^ ± 0.77
	Dehydrogense (µg TPF g^−1^ dw h^−1^)	68.45 ^b^ ± 4.11	96.65 ^a^ ± 3.26	26.76 ^d^ ± 2.26	54.55 ^c^ ± 2.65
Basic physiological parameters	Root length (cm)	23.4 ^a^ ± 1.81	25.2 ^a^ ± 1.01	12 ^c^ ± 1.87	17.4 ^b^ ± 2.07
	Shoot length (cm)	82.8 ^a^ ± 6.05	4.8 ^a^ ± 4.03	31 ^c^ ± 2.64	59.6 ^b^ ± 3.28
	Fresh weight (mg)				
	-Root	90.23 ^a^ ± 9.6	92.23 ^a^ ± 6.2	45.12 ^c^ ± 4.7	65.23 ^b^ ± 8.9
	-Shoot	252.1 ^a^ ± 13.2	264.1 ^a^ ± 10.2	151.23 ^c^ ± 9.8	198.87 ^b^ ± 12.4
	Dry weight (mg)				
	-Root	41.05 ^a^ ± 2.6	43.05 ^a^ ± 3.31	23.2 ^c^ ± 1.7	32.01 ^b^ ± 4.7
	-Shoot	95.49 ^a^ ± 5.2	99.49 ^a^ ± 1.02	58.21 ^c^ ± 3.8	65.23 ^b^ ± 8.9
	Membrane stability index (%)	72.66 ^a^ ± 2.5	75.06 ^a^ ± 2.5	50 ^c^ ± 4.0	65 ^b^ ± 1.0
	Total chlorophyll (mg g^−1^ FM)	55.4 ^a^ ± 3.35	57.5 ^a^ ± 2.05	27.2 ^c^ ± 2.41	38.78 ^b^ ± 0.34
F accumulation	Root (ppm g^−1^ DM)	Not analyzed	Not analyzed	30 ^a^ ± 3.65	20 ^b^ ± 3.71
	Shoot (ppm g^−1^ DM)	Not analyzed	Not analyzed	22 ^a^ ± 2.75	14.24 ^b^ ± 3.02
	Leaves (ppm g^−1^ DM)	Not analyzed	Not analyzed	9 ^a^ ± 1.54	4.71 ^b^ ± 1.13
	Grain (ppm g^−1^ DM)	Not analyzed	Not analyzed	2.9 ^a^ ± 0.12	0.74 ^b^ ± 0.05
Yield attributes	Panicle length (cm)	21.8 ^a^ ± 1.09	22.4 ^a^ ± 0.06	14.2 ^b^ ± 1.4	21.4 ^a^ ± 0.89
	Number of spikelets per panicle	17 ^b^ ± 1.6	20 ^a^ ± 1.0	8 ^b^ ± 1.14	15 ^b^ ± 1.3
	Number of filled grain per panicle	95 ^b^ ± 1.8	107 ^a^ ± 1.2	26 ^d^ ± 3.6	79 ^c^ ± 6.9
	Number of empty grains per panicle	5 ^c^ ± 1.3	3 ^c^ ± 0.3	41 ^a^ ± 2.3	18 ^b^ ± 1.6
	Grain length (cm)	0.8 ^a^ ± 0.04	0.8 ^a^ ± 0.02	0.5 ^b^ ± 0.05	0.7 ^a^ ± 0.05
	Grain breadth (cm)	0.26 ^a^ ± 0.01	0.26 ^a^ ± 0.0	0.15 ^c^ ± 0.01	0.2 ^b^ ± 0.0
	1000 grain weight (g)	26.8 ^b^ ± 1.3	28.9 ^a^ ± 1.1	20.4 ^c^ ± 1.14	26.8 ^b^ ± 1.4
Nutrient Contents	Protein (µg mL^−1^)	80.03 ^a^ ± 0.73	81.04 ^a^ ± 0.81	78.5 ^a^ ± 1.8	79.88 ^a^ ± 2.78
	Total sugar (µg mL^−1^)	836.6 ^a^ ± 3.2	840.6 ^a^ ± 2.7	581.4 ^c^ ± 6.7	690.9 ^b^ ± 3.7
	Iron (ppm)	43.4 ^a^ ± 1.5	45.1 ^a^ ± 1.03	33.6 ^c^ ± 1.2	37.2 ^b^ ± 1.6
	Zinc (ppm)	38 ^a^ ± 3.78	40 ^a^ ± 2.08	24 ^b^ ± 4.58	34 ^a^ ± 4.04

**Table 2 plants-14-01223-t002:** Plant growth-promoting activities of *P. aeruginosa* both in absence and presence of fluoride. Data presented are means of three replicates ± SD. Different lowercase superscripted letters (a, b) indicate statistically significant differences among treatments at *p* < 0.05, as determined by one-way ANOVA followed by DMRT post hoc test.

PGPR Traits	Without F	With F (60 mM)
Ammonia production (µg mL^−1^)	4.9 ^a^ ± 0.9	4.1 ^a^ ± 1.1
HCN production	+	+
Phosphate solubilization (µg mL^−1^)	44.93 ^a^ ± 1.3	44.53 ^a^ ± 1.6
Siderophore production index	1.10 ^a^ ± 0.1	1.23 ^a^ ± 0.2
IAA production (µg mL^−1^)	18.60 ^a^ ± 0.62	15.95 ^b^ ± 0.52
Exopolysaccharide production (µg mL^−1^)	17 ^b^ ± 1.04	24 ^a^ ± 1.1

## Data Availability

Data is contained within the article.

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
