# Peer review of "Biopriming of Pseudomonas aeruginosa Abates Fluoride Toxicity in Oryza sativa L. by Restricting Fluoride Accumulation, Enhancing Antioxidative System, and Boosting Activities of Rhizospheric Enzymes"

_plants, 2025, doi:10.3390/plants14081223_

Round 1

Reviewer 1 Report

Comments and Suggestions for Authors

The manuscript discusses the isolation of PGP soil bacterium, Pseudomonas aeruginosa (MW843625) from agricultural field and its positive implications in alleviating fluoride toxicity in O. sativa with improvement in antioxidant machinery, plant attributes and promoting enzyme activities.

Biopriming is an emerging method in which seed priming with beneficial microbes promotes plant growth, yield, biotic/abiotic tolerance, and productivity. Increased research investigations suggest the beneficial impact of biopriming in the agricultural sector. However, the process needs to address multiple limitations for complete utilization in agriculturally important crops.

I have a few suggestions for the improvement of the manuscript:

Among the beneficial microbes, why only P. aeruginosa was selected for the study? What was the rationale? Explain.

Talking about seed biopriming with P. aeruginosa, what are the future outcomes of the study? Since biopriming withholds multiple challenges and has not been completely applied in agricultural fields, how could the existing limitations will be addressed? Discuss.

Considering the multiple studies on biopriming (some on O. sativa as well, Shaffique et al, 2024, Plant stress; Vaishnavi et al, 2024, Plant Science Today, among others), how can the present study contribute to improving the growth of plants, rice being a staple food crop.

Line 31-33: ….P. aeruginosa can be exploited as ideal candidate for bioremediation……? How, needs to be discussed further.

Line 145-151: the bacterium is efficient in fluoride removal at lower concentrations. While the efficiency gradually decreases at higher fluoride concentration. It suggests that microbial bioremediation may be used at a particular concentration, and offers a limited scope. While other bioremediation methods should be explored for achieving maximum contaminant removal/mitigation.

Line 414-418: Earlier it was mentioned that bacteria can tolerate fluoride upto 60 mM concentration, however, it is not mentioned in 3.2. Fluoride resistance and removal assays. Please check.

Any examples of microbial strains (apart from Pseudomonas) that are effective in fluoride removal from the soil? Discuss examples, if any reported.

Scientific names should be used consistently and likewise in the manuscript.

Line 138: Oryza sativa L. should be rewritten as O. sativa L.

Comments on the Quality of English Language

Moderate English revisions are required.

Author Response

The manuscript discusses the isolation of PGP soil bacterium, Pseudomonas aeruginosa (MW843625) from agricultural field and its positive implications in alleviating fluoride toxicity in O. sativa with improvement in antioxidant machinery, plant attributes and promoting enzyme activities.                                                                                                                                                                                                                                                                                                                                                                                                    

Biopriming is an emerging method in which seed priming with beneficial microbes promotes plant growth, yield, biotic/abiotic tolerance, and productivity. Increased research investigations suggest the beneficial impact of biopriming in the agricultural sector. However, the process needs to address multiple limitations for complete utilization in agriculturally important crops.

I have a few suggestions for the improvement of the manuscript:

  1. Among the beneficial microbes, why only P. aeruginosawas selected for the study? What was the rationale? Explain.

Author’s response: Explanation have been added in the introduction part of the revised manuscript.

  1. Talking about seed biopriming with P. aeruginosa, what are the future outcomes of the study? Since biopriming withholds multiple challenges and has not been completely applied in agricultural fields, how could the existing limitations will be addressed? Discuss.

Author’s response: Discussed in the conclusion and future prospectus sections of the revised manuscript.

  1. Considering the multiple studies on biopriming (some on O. sativaas well, Shaffique et al, 2024, Plant stress; Vaishnavi et al, 2024, Plant Science Today, among others), how can the present study contribute to improving the growth of plants, rice being a staple food crop.

Author’s response: Discussed accordingly.

  1. Line 31-33: ….P. aeruginosacan be exploited as ideal candidate for bioremediation….? How, needs to be discussed further.

Author’s response: Discussed as asked.

  1. Line 145-151: the bacterium is efficient in fluoride removal at lower concentrations. While the efficiency gradually decreases at higher fluoride concentration. It suggests that microbial bioremediation may be used at a particular concentration, and offers a limited scope. While other bioremediation methods should be explored for achieving maximum contaminant removal/mitigation.

Author’s response: Explained accordingly.

  1. Line 414-418: Earlier it was mentioned that bacteria can tolerate fluoride up to 60 mM concentration, however, it is not mentioned in 3.2. Fluoride resistance and removal assays. Please check.

Author’s response: In the conducted study, the bacterium has been tolerated fluoride concentrations up to 200 mM (MIC). However, its maximum fluoride removal efficiency from the growth media is observed at 60 mM.

  1. Any examples of microbial strains (apart from Pseudomonas) that are effective in fluoride removal from the soil? Discuss examples, if any reported.

Author’s response: Other microbial strains which have been determined to be efficient in fluoride removal from the soil are added in the introduction of manuscript.

  1. Scientific names should be used consistently and likewise in the manuscript.

Author’s response: Checked and corrected accordingly

  1. Line 138: Oryza sativa L.should be rewritten as O. sativaL.

Author’s response: Oryza sativa L. has been replaced with O. sativa L. in whole manuscript

Reviewer 2 Report

Comments and Suggestions for Authors

In this paper, biomass accumulation, nutrient absorption, and oxidative stress tolerance in plants involving antioxidative enzymes by determining MDA accumulation and ROS (O2 .-and H2O2) in Oryza sativa under F (50 ppm) stress, oxidative stress tolerance was evaluated, which is very good. However, the logic of the article is confusing, thus it need major revision.

  1. Line 40, adding "Globally, Fluoride (F) contamination affects bones, teeth, and soft tissues leading to skeletal, dental fluorosis, and has affected 200 million individuals in 25 nations, including China, India, Korea, Mexico, Kenya and Nigeria (Rawat and Bafana, 2024)", additional adding the reference "Rawat, N. and Bafana, A. Health risk modeling and risk factors of fluorosis in the fluoride endemic village of Maharashtra: A cross-sectional study. Environmental Monitoring and Assessment, 2024, 196(12), 1230."

2. In the section of Materials and Methods, the strain was not isolated and identified in this study. So, the subtitle of 3.1 should be modified as Strain tested. Additionally, only state the scientific name, deposit site, media used of the strain.

3. Combined 2.3, 2.6, 2.9

4. Combined 2.4 and 2.8, and adjust as 2.3 before original 2.3, 2.6, 2.9

5. combined 2.7 and 2.10

6. combined 2.11, 2.12, 2.13

7.  the results of gene expression analysis?

Comments on the Quality of English Language

Many sentences are too long and difficult to understand.

Author Response

In this paper, biomass accumulation, nutrient absorption, and oxidative stress tolerance in plants involving antioxidative enzymes by determining MDA accumulation and ROS (O2.- and H2O2) in Oryza sativa under F (50 ppm) stress, oxidative stress tolerance was evaluated, which is very good. However, the logic of the article is confusing, thus it need major revision.

  1. Line 40, adding "Globally, Fluoride (F) contamination affects bones, teeth, and soft tissues leading to skeletal, dental fluorosis, and has affected 200 million individuals in 25 nations, including China, India, Korea, Mexico, Kenya and Nigeria (Rawat and Bafana, 2024)", additional adding the reference "Rawat, N. and Bafana, A. Health risk modeling and risk factors of fluorosis in the fluoride endemic village of Maharashtra: A cross-sectional study. Environmental Monitoring and Assessment, 2024, 196(12), 1230."

Author’s response: As suggested by the reviewer, the suggested line is incorporated in the introduction part along with respective reference in the bibliography.

  1. In the section of Materials and Methods, the strain was not isolated and identified in this study. So, the subtitle of 3.1 should be modified as Strain tested. Additionally, only state the scientific name, deposit site, media used of the strain.

Author’s response: Edited accordingly.

  1. Combined 2.3, 2.6, 2.9

Author’s response: The sections are combined as per reviewer’s suggestion.

  1. Combined 2.4 and 2.8, and adjust as 2.3 before original 2.3, 2.6, 2.9

Author’s response: Done as suggested by the reviewer.

  1. Combined 2.7 and 2.10

Author’s response: Sections are combined.

  1. Combined 2.11, 2.12, 2.13

Author’s response: Done

  1. The results of gene expression analysis?

Author’s response: Results of antioxidative gene expression analysis (RT-PCR analysis) are incorporated in the section 2.7.

Reviewer 3 Report

Comments and Suggestions for Authors

This manuscript reported the effect of application of a PGP soil bacterium on the accumulation of fluoride, soil properties, biomass accumulation, nutrient absorption, and oxidative stress tolerance in plants involving antioxidative enzymes in a rice plant named Oryza sativa L. The results are interesting and have a value for publication. At present there are many measurements in the text, but their relationships to the elimination of fluoride toxicity in rice plant are not clear. Hence the authors should give some nice explanations in the introduction.

The PGP bacterium would have a turnover process in the soil, which would affect its function in regulating soil properties, nutrient utilization by plants and oxidative stress tolerance in plants. These would have a varying effect on the accumulation and translocation of fluoride in plants.

For the experiment regarding different fluoride doses, the no-observed effect concentrations and effective concentrations of fluoride in regulating fluoride accumulation in plants.

For all figures, I cannot check them in the text at present. 

Comments on the Quality of English Language

The text is fine.

Author Response

This manuscript reported the effect of application of a PGP soil bacterium on the accumulation of fluoride, soil properties, biomass accumulation, nutrient absorption, and oxidative stress tolerance in plants involving antioxidative enzymes in a rice plant named Oryza sativa L. The results are interesting and have a value for publication. At present there are many measurements in the text, but their relationships to the elimination of fluoride toxicity in rice plant are not clear. Hence the authors should give some nice explanations in the introduction. 

Author’s response: As asked by the reviewer, the results are explained in the introduction part.

The PGP bacterium would have a turnover process in the soil, which would affect its function in regulating soil properties, nutrient utilization by plants and oxidative stress tolerance in plants. These would have a varying effect on the accumulation and translocation of fluoride in plants.

Author’s response: Explained in the introduction part of the revised manuscript.

For the experiment regarding different fluoride doses, the no-observed effect concentrations and effective concentrations of fluoride in regulating fluoride accumulation in plants. 

Author’s response: As suggested by reviewer, the NOEC and EC of fluoride are now incorporated in the section 3.5 of materials and methods.

For all figures, I cannot check them in the text at present. 

Author’s response: All the figures are properly cited and placed within the revised manuscript as asked by the reviewer.

Round 2

Reviewer 2 Report

Comments and Suggestions for Authors

The manuscript has been greatly improved.